# Three-Dimensional Reconstruction of *Monochamus alternatus* Galleries Using CT Scans

**DOI:** 10.3390/insects13080692

**Published:** 2022-08-01

**Authors:** Huawei Yang, Shangkun Gao, Jinxing Wang, Wen Li, Qingfeng Hou, Jianfeng Qiu

**Affiliations:** 1College of Mechanical and Electrical Engineering, Shandong Agricultural University, Tai’an 271002, China; yhw105@163.com (H.Y.); jinxingw@163.com (J.W.); 2Engineering Research Center of Forest Pest Management of Shandong Province, College of Plant Protection, Shandong Agricultural University, Tai’an 271018, China; 3College of Radiology, Shandong First Medical University, Tai’an 271000, China; liwen991217@sina.com (W.L.); hougeamm@126.com (Q.H.)

**Keywords:** *Monochamus alternatus*, CT, three-dimensional reconstruction, gallery

## Abstract

**Simple Summary:**

As one of the most important boring pests in China, the longhorn beetle *Monochamus alternatus* directly damages or kills pine trees as a vector of pine wood nematode disease. The study of the distribution of *M**. alternatus* and its gallery construction behavior is key to exploring the survival and reproductive mechanisms and population variation, as well as the associated disaster prevention and control planning. In this study, CT technology was used to scan wood segments of *Pinus densiflora* harmed by *M. alternatus*, and three-dimensional reconstruction technology was used to obtain gallery models. The results showed that the gallery types of *M. alternatus* were single and most of them were C-shaped, and a few were S-shaped or Y-shaped. There was no significant difference in the structural parameters of the three galleries. Each gallery has only one larva, which does not collude with each other. CT scanning and three-dimensional reconstruction technology can realize the non-destructive identification and detection of the gallery of *M. alternatus*, which provides a new idea and method for understanding its hidden feeding behavior and control strategy.

**Abstract:**

As a vector of pine wood nematode disease, *Monochamus alternatus* is wood-boring, cryptic, and extremely difficult to control—features that have made them a disastrous problem. In this study, we present a method of scanning the galleries of *Monochamus alternatus* using CT (computed tomography) technology to obtain their systematic structure via 3D (three-dimensional) reconstruction, so as to clarify the gallery types and their structural parameters. TLC (thin-layer chromatography) scanning on wood segments damaged by *M. alternatus* was performed using a 128-row spiral CT GE Revolution EVO to obtain 64-layer CT scanned images. From the scanned images, we were able to clearly identify the beetle larvae and their galleries. The galleries were clearly delineated from the peripheral xylem, except for parts that were blocked by a frass–feces mixture, which were slightly blurred. Three-dimensional reconstruction of the galleries showed that most of the gallery types were C-shaped, and a few were S-shaped or Y-shaped. There was only one larva per gallery, and the galleries were separate. The vicinity of the entrance hole and the anterior part of the pupal chamber were blocked with a frass–feces mixture. There were no significant differences among the galleries’ parameters, such as the width of the entrance holes, tunneling depth, vertical length, blockage length and volume, total length of the galleries, and boring volume. With MIMICS (Materialise’s interactive medical image control system) image processing software, the images of each layer were made into a composite image, providing an effective way to visualize the 3D distribution of galleries. Using the methodology outlined in this study, both a single gallery structure and the spatial distribution of multiple galleries of *M. alternatus* can be shown, and the specific parameters of galleries can also be accurately calculated, which provides new ideas and methods for carrying out ecological and scientific research and precise prevention and control techniques of *M. alternatus*.

## 1. Introduction

Longhorn beetles (Cerambycidae) are an important group of wood-boring species within the Coleoptera [1,2]. In China, species such as *Monochamus alternatus, Anoplophora glabripennis, A. chinensis, Apriona swainsoni, Batocera horsfieldi, Massicus raddei*, and *Aromia bungii* cause extensive tree damage, resulting in considerable economic losses and serious threats to ecological security [3,4,5,6,7,8,9]. These longhorn beetles share the common features of cryptic habits, long and irregular generations, high fecundity, and rapid and stable population proliferations. These species are also highly adaptable, have diverse hosts (and therefore good food resources), as well as high environmental tolerance and survival rates. Moreover, they have fewer natural enemies and weak natural control than exposed defoliator pests. Despite their weak self-diffusion ability, human activities enable them to spread easily over long distances. The beetles’ activity eventually leads to the hollowing of host plant branches, weakening of host trees, or even death of the entire plant or of whole forest areas [10,11,12]. The lack of research on the biological characteristics during the hidden life stage of wood-boring insects poses difficulty in species identification, quarantine, and control.

Longhorn beetles are different from defoliators in their exposed lives and conspicuous hazard identification. Dissecting heavily infested wood is the most direct and traditional solution for longhorn beetles, but this method destroys the integrity of the wood, and is time-consuming and laborious. At present, nondestructive detection methods for wood-boring pests in the hidden life stage mainly include manual detection, sound detection, and other methods. Manual detection methods rely on the biological reactions of winged insects to physical and chemical substances, such as trap lamps and commercial insect pheromones and attractants. These methods are not only time and labor-intensive, but also fail to attract wood-boring pests living inside trees, and are greatly affected by weather [13]. The sound detection method can be used for the early detection of wood-boring pests without harming trees. Through characteristic sound signals, the species of pests can be identified, and the number of individuals estimated, so as to carry out targeted prevention and control, which greatly reduces the cost of pest detection. However, this method is limited by the activity state of borers and the sensitivity of sound signal acquisition and analysis equipment. Meanwhile, it is necessary to determine the specific physical parameters of various insects’ sounds and establish a database [14,15,16]. In addition, X-ray detection based on X-ray autoradiography can also act as an auxiliary tool for longhorn beetle quarantine in wood import and export. Through this method, longhorn beetle larvae and the amount of wood damage can be directly observed; however, equipment is too heavy to be used in the field, and X-rays pose a risk of harm to humans [17,18].

*Monochamus alternatus* (Coleoptera: Cerambycidae) is the most destructive wood-boring pest in East Asia [19]. It is mainly harmful to *Pinus massoniana, P. elliottii, P. taeda* and *P. thunbergii, Cedrus deodara, Larix gmelinii, Abies fabri, Picea asperata*, and *Malus pumila* [20]. This species directly kills the host tree via wood drilling by larvae, or indirectly kills the host plants via pine wood nematode disease transmitted by adults during feeding, which poses a major threat to forestry production and ecological security [21]. Identifying the boring characteristics of longhorn beetles in the larval stage is important for the study of their survival and reproduction. Discovering the weak link in the larval stage is key to reducing the population of adults and can effectively prevent and control pine wood nematode disease. Zhao [22] reported in detail the biological and ecological characteristics of *M. alternatus* larvae in the gallery stage, and the results showed that the gallery distribution was closely related to tree height, diameter at breast height, plant height, and trunk bark thickness. Trees with a bark thickness of 1.1–2.0 mm and 3.1–4.0 mm had the largest gallery distribution in *P. massoniana* and *P. taiwanensis*, respectively. Togashi et al. [23] artificially inoculated *M. alternatus* larvae into the wood segments of different pine species, and the results showed that *M. alternatus* larvae of the same weight could construct deeper and longer pupal chambers in the soft xylem of the tree. Gao et al. [24] obtained the species’ gallery structure parameters by dissecting heavily infested wood, and the results showed that *M. alternatus* larvae had a single gallery structure, mostly C-shaped, unconnected with each other, and blocked to varying degrees by a frass–feces mixture near the entrance holes. Gallery length was unrelated to host plant species and larvae size. However, this method is based on dissecting a large amount of damaged wood, making it labor-intensive, and the manual measurement index data is imprecise.

Internal structure models established by X-ray computed tomography (CT) and three-dimensional (3D) image reconstruction have enabled great achievements in medicine and industry [25], and have advanced the methods for the spatial localization of wood-boring beetles and the evaluation of the degree of damage in tree trunks [17,26,27]. The 128-row spiral CT GE Revolution EVO has been greatly improved in terms of temporal and spatial resolution, providing the necessary technical support for the 3D reconstruction of longhorn beetle galleries. This study aims to achieve the following objectives with this technique: (i) determine the number and distribution of *M. alternatus* galleries in the host wood segment, and whether the characteristics of insects, frass, and boring in a single gallery can be clearly distinguished by CT scanning; (ii) use 3D reconstruction technology to accurately calculate *M. alternatus* gallery structural parameters, including gallery volume, frass blockage volume, tunneling depth and inner diameter of entrance hole; (iii) provide technical support using these methods for prevention and control strategies of *M. alternatus* in the hidden wood-boring stage.

## 2. Materials and Methods

### 2.1. Scanning Wood Segments

Five dead *Pinus densiflora* segments with entrance holes or emergence holes of *M. alternatus* were randomly selected in the Sorai Mountain view area, in Tai’an city, Shandong Province. They were cut section by section and only one section from one tree was used to scanned. The parameters of five wood segments scanned were length 100 cm×diameter 17.5 cm, length 100 cm × diameter 18.2 cm, length 100 cm × diameter 16.4 cm, length 100 cm × diameter 16.1 cm, and length 100 cm × diameter 17 cm, respective. Some *M. alternatus* in the xylem of the wood segments were still in the larval stage and their galleries were incomplete. Some had emerged, and their galleries were complete; however, some of the galleries at the ends of the segments were incomplete, i.e., where the segments had been removed from the host tree.

### 2.2. Equipment and Software

TLC (thin-layer chromatography) scanning on wood segments damaged by *M. alternatus* was performed using a 128-row spiral CT GE Revolution EVO (Manufactured by GE Healthcare and provided by the CT Laboratory at the School of Radiology, Shandong First Medical University) to obtain 64-layer CT-scanned images. Scans were run at an accelerating voltage of 120 kV and a current of 300 mA, with an exposure time of 1825 ms and effective pixel sizes of 0.339844 mm. Multiple CT images were obtained by scanning different wood segments. The 3D software used in this study was Materialise Mimics Medical 21.0 (Materialise, Brussels, Belgium, provided by the School of Radiology, Shandong First Medical University, authorized). We used Adobe Photoshop CC (2019) (San Jose, California, CA, USA) for image annotation, and MS Office 2016 (Redmond, Washington, WA, USA) for text editing and generating charts.

### 2.3. Three-Dimensional Reconstruction of M. Alternatus Galleries

Three-dimensional (3D) construction of actively selecting segmentation of gallery scan images was performed by MIMICS (2019) software (Materialise, Brussels, Belgium). Some galleries were blocked by the frass–feces mixture deposited by the *M. alternatus* larvae during feeding, which made it difficult to distinguish the galleries from the xylem in CT images. Therefore, galleries were actively selected to be segmented. First, a new mask was created and the threshold interval of 0 was selected. Then, the Edit Mask function in the Segment window was used to manually select *M. alternatus* galleries in each tomographic image. Then, the Region Grow function of MIMICS was used to select galleries more accurately to obtain the gallery images after 3D reconstruction. After the above steps, the gallery structure was continuous but not smooth, this is because the noise in CT images would generate rough 3D shapes. The Smoothing function of 3D objects was used to optimize the reconstructed gallery to achieve a more realistic display effect. The trunk and the internal gallery system were reconstructed to show the spatial distribution of multiple galleries in the trunk. The Split Mask function was used to separate each gallery from the overall reconstruction portion, allowing each gallery to be viewed individually.

### 2.4. Determination of Gallery Parameters

Maximum width of the entrance hole: first, a straight line is made along the widest part of the entrance hole, which was nearly parallel to the gallery, along the entrance hole. Then, a second straight line at a 90-degree angle to the first line to connect the widest part of the entrance hole was made. The length of the line segment was the maximum width of the entrance hole. The total length of the gallery was the length from the center of the entrance hole to the center of the emergence hole. For the reconstructed gallery, the Spline function in the Analyze window was used to draw a curve along the center part of the gallery, then the center line of the gallery and its length data was obtained. The length of the line was approximately the length of the gallery.

Segmentation of blocked galleries and their lengths: blocked galleries were consistent with the reconstruction of the other galleries and the measurement of the total length of the gallery, respectively.

Boring depth: if the gallery extended into the xylem relative to the entrance hole, the plane of the gallery entrance hole and the plane with the largest invasion depth, the annual ring was as the center of the gallery invasion hole plane, then the distance between the annual ring and the invasion hole was as the radius to make a circle. It was required that the circle should pass through the position of the entrance hole as much as possible, and the arc should coincide with the edge of the wood section as much as possible, then the annual ring center of the plane closest to the center of the annual ring in the gallery part was as the center of the circle. When the circle just touches the deepest intrusion point, the radius of the circle is determined at this time, and the absolute value of the difference between the two radii was the deepest boring depth of the gallery. If the lower part of the wood segment was thick and the gallery extends outward relative to the entrance, the annual ring was as the center of the gallery intrusion hole plane, and the distance between the annual ring and the intrusion hole was as the radius to make a circle. It was required that the circle should pass through the location of the intrusion hole as much as possible, and the arc should coincide with the edge of the wood section as much as possible. Then, a circle was drawn at the end of the gallery in the lower half. When the circle and the gallery touched, the radius of the circle was determined, so that the arc coincided with the annual ring as much as possible, the absolute value of the difference between the radii of the two circles was the deepest boring depth of the gallery.

Complete gallery volume and blocked gallery volume: volume data could be directly viewed and calculated after the segmentation was completed.

### 2.5. Data Analysis and Statistics

All observations were recorded using Windows Excel 2016 (Redmond, Washington, WA, USA). Statistical analyses were performed using GraphPad Prism version 5.0 for Windows (GraphPad Software, San Diego, CA, USA). We used one-way analysis of variance (ANOVA) to assess the differences among means in parameters of *Monochamus alternatus* galleries (entrance holes width, gallery depth, vertical length, blockage length, blockage volume, total length of the gallery, boring volume). Furthermore, we executed Duncan’s new multiple range test to compare differences among treatments.

## 3. Results

### 3.1. Scanning Images of M. Alternatus Galleries

From the data files, the CT images showed that the color of the galleries was significantly darker than that of the xylem (Figure 1A,B), which indicated the unobstructed cavity part was mainly composed of air, compared with the surrounding xylem, the X-ray attenuation was less, and the transparency in the image was higher. The boundaries, boring direction, and vertical and horizontal distribution of galleries were all clear (Figure 1C,D). The galleries blocked by the frass–feces mixture were not well-distinguished (Figure 1B), which showed that the frass–feces mixture was similar in composition to the surrounding wood. The *M. alternatus* larvae in the galleries were white in the scanning image (Figure 1E–G), showing that the X-ray attenuation of the larvae was even higher than that of the xylem. In addition, their sizes were also clearly visible, as shown in Figure 1.

### 3.2. Three-Dimensional Reconstruction of Galleries

Using the CT scanning images, the galleries in wood segments were selected, segmented, and saved, then the reconstruction data of corresponding galleries were obtained. The reconstruction models and data of 43 complete galleries were obtained from five wood segments of *P. densiflora*. As shown in Figure 2, Figure 2A–D shows views of the galleries in an entire wood segment. The relative spatial position of the galleries in the wood segment is clear. The brown parts in Figure 2 are the blocked parts of the galleries. Figure 2E,F shows the reconstructed galleries, and the shapes and number of galleries were clear. Each reconstructed gallery image was obtained by adjusting the transparency of the whole section or hiding the wood segments.

### 3.3. Gallery Types of M. alternatus

The reconstructed gallery images showed that each gallery was separate, and most were C-shaped (83.72%) (Figure 3A), while the remaining galleries were either S-shaped (6.97%) (Figure 3B) or Y-shaped (9.3%) (Figure 3C), as shown in Figure 3, which indicated that the shape of *M. alternatus* galleries was simple. Only one larva was found in each gallery and did not collude with each other. The three kinds of galleries were curved near the entrance holes, and at the end of the galleries were the pupal chambers, which were significantly larger than the entrance holes. The entrance holes and emergence holes of C-shaped and Y-shaped galleries were at the same end, while those of S-shaped galleries were at opposite ends. The three gallery types were all blocked by the frass–feces mixture near the entrance holes and in the anterior parts of the pupal chambers. It could be seen that the larvae pushed out the mixture during feeding, and it was left in the galleries after pupation.

### 3.4. Three-Dimensional Reconstruction of Gallery Blockage

It could be seen from Figure 4 that the volume of the blocked part of each gallery was 7–36% of the total gallery volume. Most of the galleries had blocked parts, and few had no blocked parts. From the CT images and reconstructed models, it could be seen that the blocked parts of the galleries were mostly located near the entrance holes or in the middle of the galleries. Figure 4A,C,D,F are the four galleries obtained after three-dimensional construction, and they were all partially blocked. The shape of the galleries in Figure 4A,C,F is “C”, and the blocked part was at the beginning of the galleries; Figure 4D was an “S-shaped” gallery, and the blocking part was located in the middle of the gallery. Figure 4B,E were the images of the transverse section of the gallery X-ray computed tomography, in which the part indicated by the red arrows were the blocked parts of the galleries. Figure 4B showed that part of the gallery was blocked in this fault; while figure U showed that the gallery was totally blocked.

### 3.5. Structural Parameters of Galleries

It could be seen from Table 1 that the parameter values of S-shaped galleries were higher than those of the other galleries, but there was no significant difference between them (*F* = 0.4, *df* = 2, 42, *p* > 0.05). There was no significant difference (*p* = 0.67) in the width of the entrance holes, indicating that the head capsule width and body width of the individuals were basically the same when entering the xylem. There were no significant differences in gallery depth, vertical length, blockage length, blockage volume, or total length of the galleries (*p* > 0.05), indicating that the boring habits and behaviors of the larvae in the xylem were similar. The proportion of galleries of each type (e.g., C-shaped) that were blocked ranged from 5% to 65%, with a mean of 24.26%. There was no significant difference in boring volume among the three types of galleries (*F* = 0.42, *df* = 2,42, *p* = 0.66), indicating that the larvae’s wood consumption and living space within the xylem were relatively fixed, and were not affected by the boring direction and behaviors.

## 4. Discussion

The study of the distribution of wood-boring pests and their gallery construction behavior is key to exploring their survival and reproductive mechanisms and population variation, as well as the associated disaster prevention and control planning [22]. In this study, our CT scan results showed that the *M. alternatus* galleries were single, that is, each gallery had only one individual larva, and the galleries were independent of each other. The three-dimensional reconstruction results showed that the majority of galleries were C-shaped, which was consistent with the results obtained by the dissection of damaged wood [24]. In addition, the results of 3D reconstruction showed that a few of the *M. alternatus* galleries were S-shaped or Y-shaped, which also reflected the intuitive nature and comprehensiveness of the three-dimensional reconstruction method. Gallery structures vary, because, in the process of boring, larvae need to make a detour when meeting host plant tree knots or change their boring direction when encountering other larvae to avoid each other. Forming a unified gallery type not only feeds conveniently but also reduces unnecessary energy consumption in the region of obstacles [28]. In this study, there were no significant differences in gallery parameters such as tunneling depth and boring volume (Table 1), which may be because of the fixed energy input of larvae while boring the gallery and constructing the pupa chamber after entering the xylem. In addition, the change of boring direction may be related to the nutrients in the regions with different xylem heights or diameters [7]. Intraspecific and interspecific aggregations have been analyzed via longhorn beetle numbers and the spatial layout of galleries in a given space [29]. In this study, the whole section of damaged wood was 70 cm long and contained nearly ten *M. alternatus* galleries, indicating that the regional population of *M. alternatus* larvae was large. Previous studies have shown that M. alternatus clusters in the host tree trunk [30,31], which may indicate that longhorn beetles choose the central area to lay eggs to avoid natural enemies and human interference during laying [7]. However, the direction of boring and the distribution of galleries were coordinated and orderly and did not disturb each other. That meant when insect density reached a certain level, the movement of the aggregation distribution center was blocked (mainly in the form of intraspecies competition), leading longhorn beetles to expand their space more to areas with less competition [32]. Some studies have shown that longhorn beetle larvae have formed a special communication mechanism during their evolutionary history. The hardened body wall tissue makes a sound when rubbed against the gallery wall, and the abdominal side plate has an organ that detects and produces sounds, and this organ senses the sound of nearby longhorn beetle larvae. Using these organs, longhorn beetle larvae locate the movements of nearby larvae by means of the mechanism of sending and receiving sounds within the dark host trunk [33]. To avoid entry by individuals of the same species, the residents of the galleries use the sound warning of rodents, thus playing an important role in regulating the degree of population aggregation, so as to effectively use the space and resources of host plants [9,34].

The galleries formed by wood-boring pests, as represented by the *M. alternatus*, have been shown to be stereoscopic. Through dissection of the damaged wood, only local conditions can be observed. In this study, X-ray CT clearly identified the insect body and the boring conditions of *M. alternatus*, which enabled the complete structure and distribution of galleries to be visualized via 3D reconstruction. Continuous scanning and three-dimensional reconstruction of galleries in different life stages better reflect the beetles’ biological characteristics (e.g., food consumption, boring habit, overwintering, and pupation characteristics). In this study, one or two generations of *M. alternatus* occurred in one year [35,36]. Larvae in the xylem stage had a small boring volume (approximately 30 mm^3^), and the structure and type of galleries were relatively simple and easy to distinguish and identify through CT scanning and three-dimensional reconstruction. Therefore, the structural characteristics and spatial distribution pattern of galleries were used as identification methods for larval species. However, the generations of species such as *Apriona swainsoni*, *Batocera horsfieldi*, *Massicus raddei*, and *Aromia bungii* span 2–3 years, with longer feeding times, increased food intake, larger gallery volumes, and more complex structures [5,7,8,37]. In addition, some longhorn beetles with more generations may also have a variety of gallery types, or irregular galleries, including galleries that connect with each other, such as seen in *Xylotrechus quadripes*, which has two generations per year and a considerable overlap in generations. Within the same trunk, the borers’ ages were inconsistent, and the galleries in the trunk occurred at different times, which led to subsequent larvae entering established galleries [38]. Therefore, in future studies, sample sizes should be increased, and a database established to determine the gallery structure for each wood-boring pest species, so as to increase the identification accuracy of subsequent testing.

In the process of feeding in the galleries, the larvae move forward and push frass and feces to the rear of the galleries by the pressing action of the hardened anterior thoracic plate, dorsal and ventral step vesicles, and caudal gluteal plate [1,9]. Among longhorn beetle species, larvae have different ways of treating the frass–feces mixture in the galleries. For example, the larvae of *Anoplophora glabripennis* and *Aromia bungii* bite a circular fecal hole and push the feces out of the hole after boring a section of the gallery [5,39]. In view of the similarity between the scanning image data of galleries and wood dust and frass, the boundary between the two must be carefully observed and the galleries carefully selected to distinguish the two successfully. Our results show that in C-shaped galleries, the mixture mainly blocked the main middle section (Figure 3A), and this blockage is likely generated during the construction of the pupa chamber. However, the mixture in other parts was pushed out of the gallery, which is not completely consistent with the report by Zhao [22] that the larvae of *M. alternatus* do not push the mixture out of the host but fill the posterior segment of the gallery. We observed many frass–feces deposits outside the entrance holes, which, according to our many years of experience in forest investigation, would have been pushed out by larvae. Therefore, we conclude that the larvae push the mixture outwards while feeding before pupation, and the mixture generated during the pupal chamber construction before overwintering remains in the gallery. To ensure the stability of the population, the borers avoid predation and parasitism by natural enemies via various covert means. For example, the depth of dung beetle tunnels has been shown to be a key variable affecting the rate at which the beetles are parasitized [40]. In the interaction between longhorn beetles and their natural enemies, the depth and length of the boring larvae galleries and the blockage by the frass–feces mixture affect their interaction with natural enemies to varying degrees. In some ichneumonid species, females laid eggs directly on host larvae by puncturing branches with their long ovipositor sheaths, and the boring depth of host larvae was a key factor affecting parasitism [41]. *Sclerodermus* spp. is an important parasitic pest group of longhorn beetles; the adult females penetrate the bark to search within the phloem for larvae to parasitize but cannot enter the xylem because of the blockage of the frass–feces mixture [34]. Before the pupal chamber is constructed by the mature larvae, the fibers bitten by the upper jaw are thick and the gap is relatively large, which provides an opportunity for their smaller natural enemies to successfully colonize or prey on them via the otherwise blocked galleries. The widely used parasitoids of Coleoptera effectively parasitize mature larvae or pupae of *M. alternatus* after overwintering [42,43,44,45], probably because the newly hatched larvae (less than 1 mm in length) can penetrate the blocked galleries to complete their parasitism.

The prevention and control of wood-boring pests is an important problem faced by researchers all over the world. At present, CT scanning and 3D reconstruction technology enable the nondestructive detection and identification of such pests, and the degree of damage by pests can be assessed through image processing. However, because of the diversity of pest species and the complexity of the harm they cause to host plants, it is still necessary to expand the amount of data collection and establish a database to ensure the detection accuracy within ecological niches and among similar species. Three-dimensional scanning of the reconstructed gallery structure provides new ideas and methods for studying the interaction between longhorn beetles and exogenous substances in xylem by simulating the gallery environment. In addition, this technology also has great application potential in pest quarantine, the use of exogenous agents, the release of natural enemies, and the research and development of related equipment for the precise control of borer pests.

## Figures and Tables

**Figure 1 insects-13-00692-f001:**
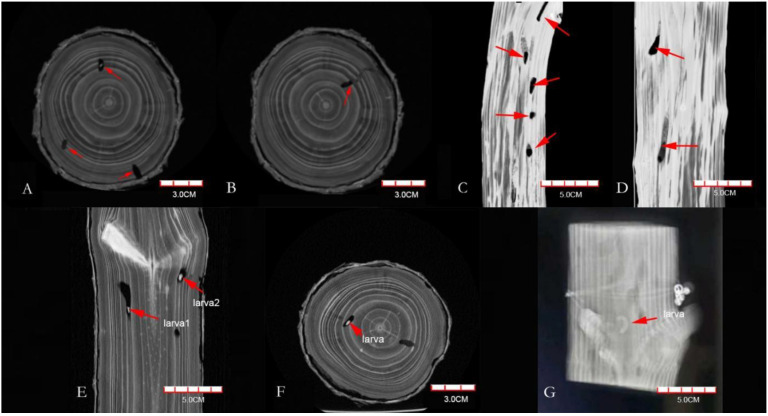
CT scanning images of *Pinus densiflora* segments ((**A**,**B**) are on the transverse plane, (**C**,**D**) are on the longitudinal plane, and (**E**–**G**) are *Monochamus alternatus* larvae (red arrow)).

**Figure 2 insects-13-00692-f002:**
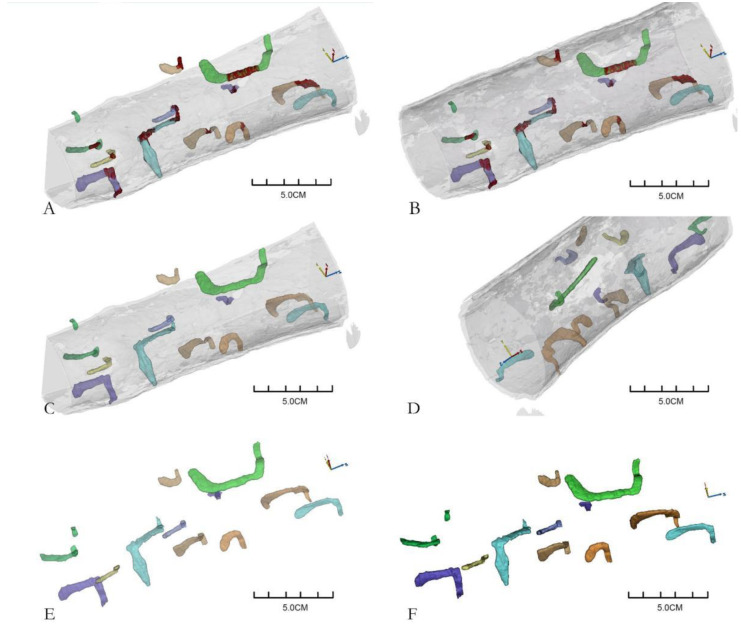
Three-dimensional reconstruction of *Monochamus alternatus* gallery structures ((**A**–**D**) are different perspectives of the three-dimensional reconstruction galleries in the same wood segment; (**E**,**F**) are two perspectives of segmented galleries. Different colors represent different galleries, except that the brown indicates the blocked part of the galleries).

**Figure 3 insects-13-00692-f003:**
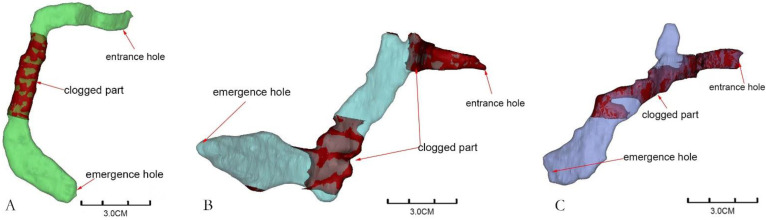
Gallery types of *Monochamus alternatus* ((**A**): C-shaped, (**B**): S-shaped, (**C**): Y-shaped).

**Figure 4 insects-13-00692-f004:**
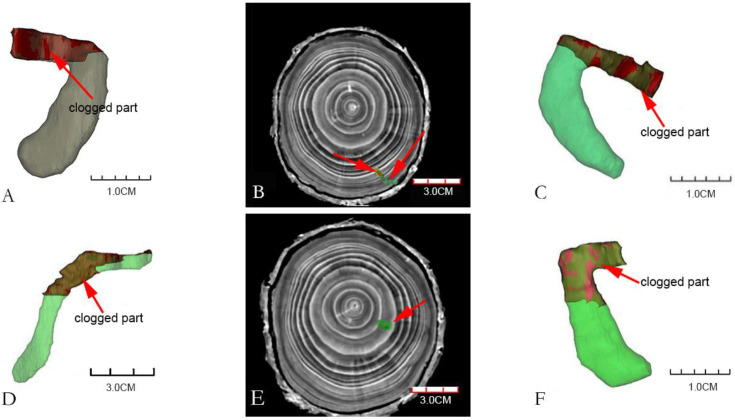
Blocked parts of *Monochamus alternatus* galleries. ((**A**,**C**,**F**): the blocked parts of C-shaped are at the beginning of the galleries; (**D**): the blocked part of S-shaped is located in the middle of the gallery. (**B**,**E**): the images of the transverse section of the CT images of the galleries, note the degree of blockage of the galleries by frass–feces mixture (red arrow), the blocked part of the gallery (indicated by green), the unblocked part of the gallery (indicated by yellow)).

**Table 1 insects-13-00692-t001:** Parameters of different gallery types of *Monochamus alternatus*. Data in the table refer to mean ± SE. The same letter within a column indicates no significant differences among treatments (*p* > 0.05).

GalleriesType	Entrance Holes Width/mm	Gallery Depth/mm	Vertical Length/mm	Blockage Length/mm	Blockage Volume/mm^3^	Total Length of the Gallery/mm	Boring Volume/mm^3^
C	2.90 ± 0.44 a	29.51 ± 8.90 a	47.68 ± 16.78 a	25.73 ± 17.10 a	806.59 ± 777.51a	79.16 ± 25.77 a	2988.01 ± 274.9 a
S	3.10 ± 0.26 a	36.44 ± 5.93 a	51.10 ± 16.48 a	38.29 ± 8.69 a	952.73 ± 296.72 a	108.73 ± 17.35 a	3669.26 ± 794.4 a
Y	2.81 ± 0.27 a	33.11 ± 10.17 a	42.60 ± 10.10 a	30.52 ± 21.84 a	558.21 ± 423.15 a	88.49 ± 29.04 a	2530.80 ± 705 a

## Data Availability

The datasets generated during and analyzed during the current study are provided in the article here.

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
