# Peer review of "Three-Dimensional Reconstruction of Monochamus alternatus Galleries Using CT Scans"

_insects, 2022, doi:10.3390/insects13080692_

Round 1
Reviewer 1 Report
This paper describes some interesting observations on the activities of M. alternatus from CT scans of Pinus wood, in particular descriptions and visual representations of the gallery shapes. It is interesting to observe the galleries in the figures, but I found the paper poorly written in places and rather frustrating to read - it was often unclear what findings were actually 'new' in this study.
Much of my frustration stems from the fact that the authors have taken a lot of measurements on the galleries they observed, but have provided limited quantitative data and have not described the statistical analysis they undertook (or really discussed what it showed. What statistical analysis was done, and what does the data in Table 1 tell the reader?).
In contrast, the discussion section is long (in some very long paragraphs which make it rather dense to read) but read much more like an introduction, and I struggled to find what was new information from this study within it and what just supported other publications (or was just published elsewhere but wasn't directly relevant). It would also be interesting to know what else could be inferred from the analysis. Some wood borers have preferences for e.g. early vs late wood - was this observed here? How close do the galleries get to another one before the larvae change direction to avoid entering another one? (That was the question that really intrigued me!) Does this study add anything to Yang's 2010 study on preference for different xylem diameters? Are these gallery shapes routinely described in this manner, do other species produce similar distinct shapes, and if so, in the same proportion (e.g. predominantly C-shaped) as these wood-borers?
The authors also end both their abstract and the discussion section with what seems like a very broad, but ultimately vague, statement of how useful the data they produce is (in terms of great promise for carrying out more ecological and scientific research and production of a 'database' of similar data) without actually providing much detail of what this would achieve or who would do it. How comparable are the results of this study to other wood-boring species? Could they be used to interpret/predict the activity of other species?
The authors need to refer back to the journal's guidelines for authors, to ensure that the sections are named and labelled correctly (the Introduction section should not be number '0'). There is no conclusions section, so the paper ends rather abruptly. I have not checked that all references in the text are listed in the reference list and vice versa, as the references need to be checked by the authors first. Some references are listed twice, some don't have complete authors named, doi's are missing (which are recommended by the journal, although not compulsory). The reference list does not need to be numbered as the references are listed by author surname alphabetically and are not numbered in the text.
Are the keywords appropriate?
I do wonder if the authors have missed some other literature that would be relevant, or which would put their work into a wider context of wood-borer study? I appreciate it might be easy to end up with a never-ending list of references, but there are some good papers on micro-CT analysis of other wood-borers that could be of interest to the authors and their readers, e.g.
Jennings & Austin (2011): Australian Journal of Entomology 50:160-163
Lehman et al (2021): J. Insect Physiology 130:104199
Brock et al (2022): Environmental Archaeology
The methods section needs to be re-written, to start with the materials, then the scanning, and then details of the software and reconstruction and parameter measurements, followed by the statistical analysis. There is currently some repetition. How were the wood samples available? Were they freshly cut down, or from the forest floor? Were they all from the same tree or several trees? How many segments were scanned?
Details of the instruments should not be in the abstract. TLC is not defined. Ensure the manufacturer's name and supplier is provided for instruments and software (e.g. 'RevolutionTM EVO Gen CT (GE Healthcare)'). Should analytical parameters, working conditions etc be included, even if just stating 'default settings were used' or similar? (I.e. Could the reader replicate the analysis from the information available in this methods section if they had the same instrument available?) Avoid writing in the first person - the methods section should not say 'we did this' as it currently does. It would be helpful to have an annotated figure to demonstrate where/how the parameters were measured. Much of the methods section is rather pedestrian and basic.
Section 2.1 is part of the results, and should not therefore repeat any methodology. The file type (dicom) should be mentioned in the method, if it is necessary anywhere, not the results section. It is not necessary to make statements like 'files were opened with' - just state the software used in the methods.
The figures are nice, in particular the representations of the galleries, but need to be presented more neatly. Ensure that individual images are attached together to avoid white spaces between them (e.g. between C, D and G). Make sure that lettering is the same size across each figure ('G' appears much bigger than 'B', for example). It is more usual to label individual images within each figure as e.g. Figure 1a, Figure 2b, rather than being labelled A-V across all 4 figures. The legend for Figure 1 needs changing as E, F and G are not larvae as currently stated, they are scans of wood which include larvae. Is it always clear what the red arrows point to? Are they large enough? What do the different colours in Figs 2, 3 and 4 represent?
The English needs improving in places:
Line 30: try to remain consistent between longicorn and longhorn
Lines 41-60 seem a bit lacking in focus
Line 67: try to be more formal than 'figuring out'. How about identifying, or understanding?
Line 86: should be 'computed' not 'computer'
Line 113: "conducted" (not "conduced")
Line 223 doesn't make sense
Line 259: mm3 needs to be superscript (the 3 only)
Line 334: "are provided in"
Author Response
This paper describes some interesting observations on the activities of M. alternatus from CT scans of Pinus wood, in particular descriptions and visual representations of the gallery shapes. It is interesting to observe the galleries in the figures, but I found the paper poorly written in places and rather frustrating to read - it was often unclear what findings were actually 'new' in this study.
We have revised and improved some parts of the manuscript again.
Much of my frustration stems from the fact that the authors have taken a lot of measurements on the galleries they observed, but have provided limited quantitative data and have not described the statistical analysis they undertook (or really discussed what it showed. What statistical analysis was done, and what does the data in Table 1 tell the reader?).
We rewrote statistical analysis and a part of result.
In contrast, the discussion section is long (in some very long paragraphs which make it rather dense to read) but read much more like an introduction, and I struggled to find what was new information from this study within it and what just supported other publications (or was just published elsewhere but wasn't directly relevant). It would also be interesting to know what else could be inferred from the analysis. Some wood borers have preferences for e.g. early vs late wood - was this observed here? How close do the galleries get to another one before the larvae change direction to avoid entering another one? (That was the question that really intrigued me!) Does this study add anything to Yang's 2010 study on preference for different xylem diameters? Are these gallery shapes routinely described in this manner, do other species produce similar distinct shapes, and if so, in the same proportion (e.g. predominantly C-shaped) as these wood-borers?
Yes, some wood borers have preferences for e.g. early vs late wood. But the wood borer Monochamus alternatus has preference for weakened pines. The larva of M. alternatus would not enter into xylem until three instar, once enters, it will feed a large amount of wood xylem with clear biting sound that is enough to be heard by nearby larvae. If two larvae meet in a gallery, they will kill each other by mandibles, which is not conducive to maintaining their population. Generally, the number of larvae on a wood segment is controlled by the oviposition of adults. Even if the larvae gather in a host tree, their galleries will not connect with each other. At least we didn't find connected galleries through anatomical wood segment and CT scanning. Different xylem diameters (Yang, 2010) may affect the parameters of the galleries, but will not affect the shapes of galleries. So far, no other longhorn beetle has been found to feed this shape of gallery. Gao et al., (2022) found that the C-shaped gallery of M. alternatus by Artificial dissection of damaged pine segments. We found other shapes with small proportion e.g. S-shaped, Y-shaped, by CT scanning, which are variants of form C-shaped.
The authors also end both their abstract and the discussion section with what seems like a very broad, but ultimately vague, statement of how useful the data they produce is (in terms of great promise for carrying out more ecological and scientific research and production of a 'database' of similar data) without actually providing much detail of what this would achieve or who would do it. How comparable are the results of this study to other wood-boring species? Could they be used to interpret/predict the activity of other species?
Because of the hidden life of borers, people can not understand their activities and behaviors in the xylem without damage, and even the most basic gallery shapes and boring direction are not easy to judge, so it is difficult to control borers in China. In this study, we can clearly see their internal activities through CT scanning, while do not damage the host plant, so we have the above discussion and prediction. In future, we will collect more galleries of various longhorn beetles by CT scanning to predict the activity of other species.
The authors need to refer back to the journal's guidelines for authors, to ensure that the sections are named and labelled correctly (the Introduction section should not be number '0'). There is no conclusions section, so the paper ends rather abruptly. I have not checked that all references in the text are listed in the reference list and vice versa, as the references need to be checked by the authors first. Some references are listed twice, some don't have complete authors named, doi's are missing (which are recommended by the journal, although not compulsory). The reference list does not need to be numbered as the references are listed by author surname alphabetically and are not numbered in the text.
We have checked all references in the text and added the complete authors named or doi’s.
Are the keywords appropriate?
We replaced the keywords with Monochamus alternatus; CT; three-dimensional reconstruction; gallery;
I do wonder if the authors have missed some other literature that would be relevant, or which would put their work into a wider context of wood-borer study? I appreciate it might be easy to end up with a never-ending list of references, but there are some good papers on micro-CT analysis of other wood-borers that could be of interest to the authors and their readers, e.g.
Jennings & Austin (2011): Australian Journal of Entomology 50:160-163
Lehman et al (2021): J. Insect Physiology 130:104199
Brock et al (2022): Environmental Archaeology
Thanks for your good papers. We are very interested in micro CT and have read many related papers before. But there is a big difference between micro CT and 64-layerCT in the scope and method of use.
The methods section needs to be re-written, to start with the materials, then the scanning, and then details of the software and reconstruction and parameter measurements, followed by the statistical analysis. There is currently some repetition. How were the wood samples available? Were they freshly cut down, or from the forest floor? Were they all from the same tree or several trees? How many segments were scanned?
Five dead Pinus densiflora segments with entrance holes or emergence holes of M. alternatus were randomly selected in the Sorai Mountain view area, in Tai’an city, Shandong Province. They were cut section by section and only one section from one tree was used to scanned. The parameters of five wood segments scanned were length 100cm×diameter 17.5cm, length 100cm×diameter 18.2cm, length 100cm×diameter 16.4cm, length 100cm×diameter 16.1cm and length 100cm×diameter 17cm, respective.
Details of the instruments should not be in the abstract. TLC is not defined. Ensure the manufacturer's name and supplier is provided for instruments and software (e.g. 'RevolutionTM EVO Gen CT (GE Healthcare)'). Should analytical parameters, working conditions etc be included, even if just stating 'default settings were used' or similar? (I.e. Could the reader replicate the analysis from the information available in this methods section if they had the same instrument available?) Avoid writing in the first person - the methods section should not say 'we did this' as it currently does. It would be helpful to have an annotated figure to demonstrate where/how the parameters were measured. Much of the methods section is rather pedestrian and basic.
TLC (Thin-Layer Chromatography) scanning on wood segments damaged by M. alternatus was performed using a 128-row spiral CT GE Revolution EVO (Manufactured by GE Healthcare and provided by the CT Laboratory at the School of Radiology, Shandong First Medical University) to obtain 64-layer CT scanned images. Scans were run at an accelerating voltage of 120kV and a current 300 mA, with detector width 40 mm, pitch 0.516 mm, speed 120.62 mm/rotation, layer thickness 0.625 mm and rotation time 1.0 s. Multiple CT images were obtained by scanning different wood segments.
Section 2.1 is part of the results, and should not therefore repeat any methodology. The file type (dicom) should be mentioned in the method, if it is necessary anywhere, not the results section. It is not necessary to make statements like 'files were opened with' - just state the software used in the methods.
We have revised it.
The figures are nice, in particular the representations of the galleries, but need to be presented more neatly. Ensure that individual images are attached together to avoid white spaces between them (e.g. between C, D and G). Make sure that lettering is the same size across each figure ('G' appears much bigger than 'B', for example). It is more usual to label individual images within each figure as e.g. Figure 1a, Figure 2b, rather than being labelled A-V across all 4 figures. The legend for Figure 1 needs changing as E, F and G are not larvae as currently stated, they are scans of wood which include larvae. Is it always clear what the red arrows point to? Are they large enough? What do the different colours in Figs 2, 3 and 4 represent?
We readjusted the pictures.
The English needs improving in places:
Line 30: try to remain consistent between longicorn and longhorn
We have replaced longicorn with longhorn in the whole text.
Lines 41-60 seem a bit lacking in focus
Maybe, so we added a sentence “Longhorn beetles are different from defoliators in their exposed lives and conspicuous hazard identification”.
Line 67: try to be more formal than 'figuring out'. How about identifying, or understanding?
We have replaced “figuring out” with “identifying”.
Line 86: should be 'computed' not 'computer'
We have replaced “computer” with “computed”.
Line 113: "conducted" (not "conduced")
We have revised it.
Line 223 doesn't make sense
We have rewritten this sentence “In this study, the whole section of damaged wood was 70 cm long and contained nearly ten M. alternatus galleries, indicating that the regional population of M. alternatus larvae was large.”
Line 259: mm3 needs to be superscript (the 3 only)
We have revised it.
Line 334: "are provided in"
We have revised it.
Reviewer 2 Report
Comments attached as a PDF.

Author Response
Comments
General comments: The paper is well-written, employs a sound methodology and provides clear and interesting results. The statistics used need to stated clearly otherwise it is not possible to validate the accuracy of the statistical analyses. The results are displayed clearly but some improvements could be made to the figures to improve their clarity and impact (as described below). Some of the descriptions of the X-ray CT methodology and analysis would benefit from some clarification as the wrong terms are used or the most useful information is sometimes missing. This paper presents high quality images and segmentations of bore holes of M. alternatus which are clearly useful in understanding the biology and impact of this species. The method of segmenting bore holes and producing 2D and 3D measurements of them has been rarely done so this paper definitely contributes to the field. However, there are quite a few papers with CT scans of bore holes, therefore it would have been a great opportunity to present bore holes from different species or the same species in different hosts, to provide some comparisons that are missing from the literature and would make this paper more novel.
Thank you for your recognition of this study.
Abstract: The English in the abstract needs some correction, and the abstract itself could be written more concisely. Some terms are used incorrectly: ‘CT technology’ when CT hasn’t been defined yet, use ‘X-ray computed tomography’. ’64-layer CT scanned images’ not clear what 64-layer means, possibly 64-bit? Also remove the word ‘scanned’. ‘TLC’ is used here and not defined, also it isn’t used or defined in any other part of the article. The last sentence “which provides new ideas and methods for carrying out ecological and scientific research and precise prevention and control techniques of M. alternatus” is very vague, if it provides new ideas/methods then you can mention at least one of them.
L30: I think it is better to stick to one common name descriptor, longicorn is used many times in the introduction but throughout the rest of article longhorn is used.
We have replaced longicorn with longhorn in the whole text.
L75: Not sure how the larvae warning sound is related to the ideas being discussed in this paragraph.
We have deleted this sentence and put it in the discussion.
L86: Computed tomography not ‘computer tomography’.
We have revised it.
L89: You should cite ‘Burrow forms, growth rates and feeding rates of wood-boring Xylophagaidae bivalves revealed by micro-computed tomography by Amon et al, 2015’ here as well, as it is the first study to segment bored holes in wood as was done here. Also, I would recommend checking the literature for more references as I found a number of other papers that employ the same approach as used here to investigate wood-boring animals.
Yes, there were a lot of good papers and approaches to investigate wood-boring animals, but the CT used in our study was different from micro-CT, e.g. scope of application, body size of the detection object and method of application.
Section 1.2: What statistics software was used? And what statistical tests?
We have added the part “2.5 Data analysis and statistics
All observations were recorded using Windows Excel 2016. We used one-way analysis of variance (ANOVA) to assess the differences among means in parameters of Monochamus alternatus galleries (entrance holes width, gallery depth, vertical length, blockage length, blockage volume, total length of the gallery, boring volume). Furthermore, we executed Duncan's new multiple range test to compare differences among treatments.”
Section 1.3: Vital scanning parameters are missing (voxel size i.e. resolution, exposure time, no. of X-ray projections) and some stated are not useful (speed and rotation time).
L124: Should be ‘actively selected to be segmented’ not reconstructed.
We have revised it.
L130: ‘a more realistic display effect’ is not very scientific reason for smoothing, I suspect that your aim by smoothing was to make the segmentation more accurate as noise in the CT images would produce a rougher/less realistic 3D shape without smoothing.
Measurements: Was each gallery measurement (diameters and lengths) just made once per gallery? There is error associated with manual segmentation (different parameters of the region growing tool or type/degree of smoothing applied) which will produce changes in the 3D shape and therefore the measurements made. This could cause under or overestimation of measurements so it would be good to have a validation of the results using different segmentations (e.g. different users or slightly different parameters) so we know how much error is associated with these measurements, in addition to the error produced by the voxel size which should be mentioned too. Also, Boring depth measurements are not clearly stated how they were made, it is a bit confusing as is currently written.
Yes, thank you for your professional suggest. Although each gallery measurement was just made once per gallery, the measurement of 43 galleries were made in our study. Maybe there were some error associated with manual segmentation, we hoped to reduce them by more repeats and qualitatively described the type of gallery in our study.
L157: Segmentation not reconstruction.
We have revised it.
L162: Instead of referring to the colour being darker, you should refer to the X-ray attenuation (as that is what the colour represents). Obviously the galleries (being composed mostly of air) will have less X-ray attenuation than the surrounding wood, which gives the colour difference.
We rewrote the sentence “From the data files, the CT images showed that the color of the galleries was significantly darker than that of the xylem (Figture 1A, 1B), which indicated the unobstructed cavity part was mainly composed of air, compared with the surrounding xylem, the X-ray attenuation was less, and the transparency in the image was higher. The boundaries, boring direction, vertical and horizontal distribution of galleries were all clear (Figture 1C, 1D). The galleries blocked by frass-feces mixture were not well-distinguished (Figture 1B), which showed that frass-feces mixture has similar x-ray attenuation to the surrounding wood. The M. alternatus larvae in the galleries were white in the scanning image (Figture 1E, 1F, 1G), showing that the X-ray attenuation of larvae was even higher than that of xylem, but lower than that of empty gallery. In addition, their sizes were also clearly visible, as shown in Figure 1.”.
L164: Not clear what is meant by “The galleries blocked by frass-feces mixture resemble xylem and were not well-distinguished.” Presumably you mean that the frass-feces mixture has similar x-ray attenuation to the xylem.
Yes, we have advised the sentence.
L165: Again referring to colour here is not describing what is actually happening (with regard to X-ray attenuation), also it is not clear what is meant by “but were conspicuous in the dark galleries”. From your figures I can see that the larvae have a higher x-ray attenuation than even the xylem, so there would be a clear contrast between the larvae and the empty gallery.
We rewrote the sentence “The M. alternatus larvae in the galleries were white in the scanning image (Figture 1E, 1F, 1G), showing that the X-ray attenuation of larvae was even higher than that of xylem, but lower than that of empty gallery.”
L169: It is important that throughout the article you are clear about the definition of CT terms. Here and in multiple other places you state that the galleries are reconstructed, but reconstruction is the process that happens to all the X-ray projections collected to produce the tomographic slice data. I suspect that in most cases you mean to say segmented instead of reconstructed. As this is the process by which you separate the galleries from the surrounding xylem by assigning them the false colours/masks.
Thanks very much for your great recommend, we have replaced “reconstructed” with “segmented”.
L171: This is confusingly phrased. “data of 43 complete galleries were obtained by reconstructing five complete galleries in P. densiflora”, it cannot be possible to obtain 43 complete galleries from 5 complete galleries, so needs rephrasing so that it’s clear what you mean.
This sentence has been revised “The reconstruction models and data of 43 complete galleries were obtained from five wood segments of P. densiflora.”
L172: I think you mean to make the point that once segmented the 3D model can be easily manipulated to perform measurements, think as it is written the significance of this sentence is missing.
We have deleted this sentence.
L175: You need to make sure to state that each section (H,I,J,K) is part of figure 2 as that is not clearly stated in this paragraph. Also make sure to maintain the correct verb tense in this paragraph (are not were) as these are the results you have and are describing.
We have re-described the results “Figure 2. Three-dimensional reconstruction of Monochamus alternatus gallery structures (2A, 2B, 2C, 2D, are different perspectives of the three-dimensional reconstruction galleries in the same wood segment; 2E ,2F are two perspectives of segmented galleries).”
L176: You mention that brown represents the blocked parts in fig. 2H but neither here nor in the figure label do you mention what all the colours represent and for all parts of the figure.
We have added description in the figure label.
L192: The ‘0-40 percent’ seems too neat, also states that this can be seen from figure 4, please calculate this value instead (should be easy if it’s already segmented as it appears to be from fig. 4) and provide specific values.
We have advised this part “It could be seen from Figure 4 that the volume of the blocked part of each gallery was 7%–36% of the total gallery volume”.
Section 2.5: P values are stated but no mention made of what statistical tests were used to make those comparisons, making it impossible to know if the correct test was used and therefore if these results are valid or not.
We have advised this part “2.5 Data analysis and statistics All observations were recorded using Windows Excel 2016. Statistical analyses were performed using GraphPad Prism version 5.0 for Windows (GraphPad Software, San Diego, CA). We used one-way analysis of variance (ANOVA) to assess the differences among means in parameters of Monochamus alternatus galleries (entrance holes width, gallery depth, vertical length, blockage length, blockage volume, total length of the gallery, boring volume). Furthermore, we executed Duncan's new multiple range test to compare differences among treatments.”
L223: Not clear what is meant by “This boring behavior not only feed conveniently”.
“This boring behavior” mean “Forming an unified gallery type”.
L252: This sentence needs re-wording, in particular, change ‘CT scanning technology’ to ‘X-ray CT’ and ‘to be reconstructed via 3D reconstruction’ to ‘to be visualized via…’ or something similar.
We have re-wording this sentence, “In this study, X-ray CT clearly identified the insect body and the boring conditions of M. alternatus, which enabled the complete structure and distribution of galleries to be visualized via 3D reconstruction”.
Discussion: The final paragraph could use some more detail of potential benefits of this technique as this clearly can’t be used in the field so it seems that the aim is to replace manual dissection of wood sections, so it would be good to re-iterate the benefits of this technique versus the traditional techniques. Also mention the limitations such as that it can’t be used in the field to detect infestations in living trees.
We think that X-ray CT is not only to replace manual dissection of wood sections, but also would be used in the field to detect infestations in living trees in the future. Because we are trying to assemble a vehicle mounted X-ray CT to realize that.
Figures: The figures are of good quality but could use some cleaning up. All the images need more clear arrows (hard to see some), scalebars shouldn’t overlap with the images too much as it makes them hard to see, the scalebar labels vary in size they should be more consistent and generally more clear as some are hard to read, labels in the figures also suffer from being varied sizes and often too small, sub-figure labels (e.g. A, B, C) are written in a font that is too thin in places making I, J and L look the almost the same they need to be clearer and ideally a similar size. I think that the sub-figure labels should start at A for each figure instead of continuing from H in fig. 2. Fig. 1A + B could use more contrast as the image is a bit dark. Fig. 2 each individual image can larger (looks small compared to the other figures) and the colours needs explaining in the caption. Fig. 3 the colours need explaining in the caption. Fig. 4 I can’t see blockage in R and U, I would recommend showing the gallery as a region of interest so we can have a ‘zoomed-in’ view and therefore see the blockage more clearly. Also would be worthwhile to see a photo of the wood (ideally in a similar cross section to the CT images) so we can verify that the CT images correlate to what you see after dissection.
We have reprocessed the figures.
Table 1: The inclusion of the statistical parameters here is confusing, not least because they appear under ‘galleries type’ but also as it is not stated what statistical test was applied and what is being compared.
We have advised the label of table 1 and added the description about what statistical test was applied and what is being compared in “2.5 Data analysis and statistics”.
References: Lyons et al. and Ma et al. listed twice in the reference list with different dates but otherwise identical.
We have deleted one of them.
Round 2
Reviewer 2 Report
I would like to thank the authors for their changes and improvements to the paper, I think this paper is of a high quality and worthy of publication. I think there are just a few typographical and grammatical errors left that need correction, I have listed some here:
L154: There are a few typos in this sentence “First, a new mask was crested and the threshold interval as 0 was selected”. ‘Crested’ should be ‘created’ and ‘interval as 0’ should be ‘interval of 0’.
L155: Also here “Segment window was useded” should be ‘used’.
L172: used is misspelled.
Section 2.3 is in past tense and section 2.4 is in present tense, please make sure the correct tense is used throughout the article.
L187: Gallery is misspelled.
L210: Figure is misspelled and also through all the results section.
Author Response
Once again, we sincerely thank you for your suggestions on the manuscript. We have revised the errors or inadequacies and marked them in the manuscript. The following is the specific content.
L154:We have revised the sentences.
L155:We have revised the word.
L172:We have revised the word.
2.4:We have changed the tense in Section 2.4 to the past tense.
L187:We have revised the word.
L210:We have changed the spelling of the word "figure" in results section.